# Initial Oxidation Behavior of AlCoCrFeNi High-Entropy Coating Produced by Atmospheric Plasma Spraying in the Range of 650 °C to 1000 °C

**DOI:** 10.3390/ma17030550

**Published:** 2024-01-23

**Authors:** Rong Chen, Xin You, Ke Ren, Yuwei Liang, Taihong Huang, Biju Zheng, Peng Song

**Affiliations:** 1Faculty of Materials Science and Engineering, Kunming University of Science and Technology, Kunming 650093, China; bearclan0014@gmail.com (R.C.); w19834536658@163.com (X.Y.); 19712805560@163.com (K.R.); taihonghuang@hotmail.com (T.H.); zhengbiju@126.com (B.Z.); 2Yunnan Engineering Research Center of Metallic Powder Materials, Kunming 650093, China; 3Faculty of Civil Aviation and Aeronautics, Kunming University of Science and Technology, Kunming 650500, China

**Keywords:** plasma spraying, oxidation behavior, high-entropy coatings, phase transition

## Abstract

As protective coatings for the thermal parts of aero-engines, AlCoCrFeNi coatings have good application prospects. In this study, atmospheric plasma spraying (APS) was used to prepare AlCoCrFeNi high-entropy coatings (HECs), which were oxidized from 650 °C to 1000 °C. The mechanism of the oxide layer formation and the internal phase transition were systematically investigated. The results show that a mixed oxide scale with a laminated structure was formed at the initial stage of oxidation. The redistribution of elements and phase transition occurred in the HECs’ matrix; the BCC/B2 structure transformed to Al-Ni ordered B2 phase and Fe-Cr disordered A2 phase.

## 1. Introduction

High-entropy alloys (HEA) are a new type of advanced engineering alloy suggested by Yeh et al. and Cantor et al. in 2004 [1,2,3,4]. Unlike conventional alloys (e.g., nickel-based superalloys, stainless steels, etc.), HEAs are composed of more than five elements whose atoms are close to the iso-atomic ratio. They have high strength, good corrosion resistance, oxidation resistance and other properties at elevated temperatures. The reasons for the excellent performance of high-entropy alloys can be attributed to four major effects: thermodynamic high entropy, kinetic hysteresis diffusion, the organizational structure of the lattice distortion and the performance of the “cocktail” effect. High-entropy alloys have a high degree of structural complexity, and it is easy to form a single solid solution (single-phase FCC/BCC or two-phase FCC + BCC), which results in excellent mechanical properties. Coatings prepared from high-entropy alloys have the same excellent mechanical, high-temperature, wear-resistant and corrosion-resistant properties as thin films, and have broad application prospects in many fields. At present, great attention is being paid to this new alloy [5,6,7,8,9,10].

AlCoCrFeNi alloy is a typical type of HEA, and its oxidation resistance has been attracting much attention. It has been found that Al_2_O_3_ and Cr_2_O_3_ scales are formed on AlxCoCrFeNi alloys due to the selective oxidation of Al and Cr at high temperatures [5,11,12,13,14]. Alloys can have greater oxidation resistance if the continuity and internal position of the Al_2_O_3_ scales are improved by enhancing the content of Al [15]. Lu concluded that the oxidation of AlCoCrFeNi alloys would be slower if small amounts of Y and Hf were added [16]. Their oxidation behavior was studied by Butler and Weaver in dry air at 1050 °C [5]. It has been suggested that they are the perfect material for high-temperature constructional praxis as they have strong oxidation resistance and structural stability. Therefore, AlCoCrFeNi alloys have been proposed as high performance antioxidant coating materials for protecting the high temperature components of aero-engines or as bonded coating materials in TBCs. Cheng prepared AlCoCrFeNi coatings using plasma spraying, and their phase composition could be changed with different spraying power and plasma gas [17]. Lu pointed out that gas-atomized AlCoCrFeNi powder showed a low oxidation rate under the isothermal condition of 900–1000 °C, showing that it was a potential raw material for spraying [18]. Thus, it is significant to study the high-temperature oxidation of AlCoCrFeNi coatings to explore their practical applications.

In addition to their oxidation resistance, their phase transition is also worth investigating. Compared with other components of the AlCoCrFeNi system, this alloy has a complicated phase transition [19,20,21,22,23]. The microstructure and phase composition of the equiatomic AlCoCrFeNi have been thoroughly studied by Munitz et al. [24]. The XRD of samples experiencing quench at critical temperatures was analyzed and the following phase transitions were found in the system: BCC + B2 (650 °C) → B2 + FCC + σ (975 °C) → BCC + B2 + FCC (1200 °C) →BCC + B2 + FCC (traces) [25]. Their microstructural analyses proved this result and were in agreement with the results given by Wang et al. [26]. In Lobel’s study, a biphasic structure dominated by FCC phases was found in the AlCoCrFeNi coatings prepared using HVOF, and a more homogeneous grain size was produced after annealing at 1050 °C [27]. FCC reduced the hardness of the coatings, but it also increased their ductility, resulting in better wear resistance. In view of the fact that the different phase structures have a great influence on the mechanical properties of the coatings, it is of great significance to further investigate the phase structure of AlCoCrFeNi coatings. And the formation of phases during the oxidation process is related to different atomic size parameters. Mohsen Saboktakin Rizi [28] has already conducted in-depth research.

AlCoCrFeNi alloys can be used as protective coatings for the thermal parts of aero-engines, which gives them broad application prospects. Researchers have made a relatively systematic study of the AlCoCrFeNi alloy, but there are few studies of the AlCoCrFeNi coating, and the research into the high temperature oxidation and phase transformation of AlCoCrFeNi coatings is not comprehensive enough. This work, therefore, focuses on this aspect. The AlCoCrFeNi coatings were prepared by atmospheric plasma spraying (APS) and oxidized isothermally at 650–1000 °C in order to study their oxidation behavior [29,30,31,32]. The formation and evolution process of oxides were determined, and the phase transition of the coating interior was investigated to find out the oxidation mechanism.

## 2. Experiment Details

### 2.1. Materials and Methods

Commercial AlCoCrFeNi powders prepared by gas atomization were used as raw materials. Ni-based alloy samples were sandblasted and served as the substrate receiving spraying. The APS device Metco-F4MB-XL-6 mm (Oerlikon, Wohlen, Switzerland) was used to conduct the experiment. The parameters of the APS are listed in Table 1. H2 and Ar were used as the processing atmosphere during the spraying of the AlCoCrFeNi coatings so that the spraying power and current could be controlled. After spraying, the HECs samples were put in a tube furnace and then exposed to temperatures of 650, 800, 900, and 1000 °C for 20 h, 50 h, and 100 h, respectively. Then they were taken out and cooled until they reached room temperature.

### 2.2. Characterization

The samples were subjected to X-ray diffraction (Miniflex-600, Rigaku, Japan) and the diffraction patterns were analyzed. The measurement 2θ was used in the angular range of 10–90° with a scanning speed of 10°/min, a step size of 0.02°, and scanning intervals of 1 s/step. Scanning electron microscopy was used to characterize the microscopic morphology of different areas on the sample surface. They were inlaid by Goral cold-inlaid epoxy resin and polished by a metallographic polishing machine before meeting the standards for analysis and testing. Their cross-section morphology and element distribution were analyzed by SEM and BRUKER-X (BRUKER, Karlsruhe, Germany) flash energy dispersive spectrometer (EDS). The volume fraction of their phases was calculated by using ImageJ 1.52 v software and SEM images.

## 3. Results and Discussions

### 3.1. Characterization of the Raw Material of Powder and Sprayed Coatings

In Figure 1, the surface morphology, particle size, and phase structure of the AlCoCrFeNi powder are presented. The powder was regular and spherical shaped and had a smooth surface and uniform size (see Figure 1a,b). Figure 1c shows the particle size distribution of the powder. The particle diameters were less than 50 μm with an average particle size of 20 μm. The atomic content of each element based on EDS in a single powder particle is shown in Table 2, which showed that it was similar to the equal molar ratio. This implied that all of its elements were evenly mixed, and it could be used as spraying powder. The XRD spectra are shown in Figure 1d. The powder had a mixture structure of BCC and B2 phases (ordered BCC structure). Its main diffraction peaks were strong in intensity and narrow in width, indicating that the powder had high crystallinity and was a standard type of HEA powder. The crystallite size of the powders at different angles was 30.07 nm (θ = 15.53), 91.80 nm (θ = 22.24), 76.85 nm (θ = 32.38) and 52.22 nm (θ = 41.02). The volume fraction of the phases was 90%, 98%, 94% and 92%, respectively [33].

The surface and cross-section morphology of the sprayed HECs can be seen in Figure 2. Figure 2a,b show the surface topography, which had an overall uneven appearance typical of coatings subjected to plasma spraying. There were island-like and protruded tissues as well as smooth tissues on the surface. These island-like ones were formed by the semi-solid deposition of partially melted large-size and medium-size powder particles, while smooth ones were formed by completely melted liquid deposition. Figure 2c,d show the cross-section morphology of the coating. It can be seen that the coating had a thickness of about 250 μm and a typical layered structure. It looked even and compact, had a few pores and had experienced slight oxidation, but no cracks were observed. Some undulations could be seen on the surface. The coating was well bonded with the substrate, showing mechanical bonding. The XRD pattern in Figure 3a shows that the phase composition of the coating after spraying was consistent with that of the powder, indicating that the spraying process had not changed the phase structure of the coating.

### 3.2. Phase Transition during High-Temperature Oxidation

The black line in Figure 3a represents the XRD image of the coating after spraying, which is the same as the XRD pattern of the powder. This implies that the coating still has a mixture of BCC and B2 phases after the spraying, i.e., plasma spraying does not change its phase structure. The XRD of the sample oxidized at 650 °C for 200 h (red line in Figure 3a) showed peaks of the FCC phase and σ phase in addition to the BCC/B2 peaks. σ phases also appeared, and BCC phases may transform into σ phases at a high temperature [24]. It has been reported that σ-phase precipitation occurs in the temperature range of 650–800 °C, and it is dissolved at temperatures between 850 °C and 900 °C [24,34]. The σ-phase peaks of XRD (blue line in Figure 3a) almost disappeared when oxidized at 800 °C for 100 h, while the σ-phase peaks of XRD (green line in Figure 3a) completely disappeared when oxidized at 900 °C for 100 h. This is in line with the results given by Wang et al. [26]. Significant BCC–FCC phase transition occurred after oxidation, which is related to the formation of oxides on the surface and the mobility of Al. The peak of Al_2_O_3_ appeared after the coating was oxidized at 800 °C for 100 h, which can be explained by the selective oxidation of Al under an increased temperature, leading to a rapid growth of Al_2_O_3_ on the coating surface. Figure 3b shows the XRD images of the coating oxidized at 1000 °C for different time frames. After 20 h of oxidation, the coatings were stabilized, with a three-phase structure of BCC-FCC-Al_2_O_3_.

### 3.3. Observation of the Microstructure

Figure 4 is the SEM image of the coating surface. After oxidation at 650 °C, it shows the typical surface morphology, which is almost the same as that of other coatings after receiving spraying. The morphology had an island-like convex surface and smooth bottom, and almost no oxide was found. The surface morphology of the coating after being oxidized at 800 °C is shown in Figure 4d–f. Compared with the surface morphology at 650 °C, it was found that there were some particles, especially on its smooth bottom surface, and the number of particles increased significantly with the oxidation time, as shown in points 1 and 2 in Table 3, which was the result of EDS. Most of the oxides were Al_2_O_3_ and spinel oxides, which is consistent with the results given by Zhang [34]. Figure 4g–i shows the surface morphology of the coating oxidized at 900 °C, which was significantly different from that of the coating oxidized at 800 °C. A dense oxide layer was formed there, which was dominated by α-Al_2_O_3_. After oxidation for 20 h (see Figure 4g), many needle-like oxides were seen on the surface, which gradually decreased after oxidation for 50 h and disappeared after oxidation for 100 h. The needle-like oxides were identified as θ-Al_2_O_3_ [34], and their gradual disappearance proves the transition from θ-Al_2_O_3_ to α-Al_2_O_3_ [35,36]. The surface morphology of the coating oxidized at 1000 °C is shown in Figure 4j–l. Compared with the surface at 900 °C, the oxide layer of this one was thicker. As can be seen from points 3 and 4, the aluminum content decreased significantly, while the iron content increased significantly.

In Figure 5a–c, the cross-section SEM image of the coating oxidized at 650 °C is presented. No traces of oxide were seen on its surface, but some were found in the lamellar gap inside the coating. On the whole, the increase in oxidation time did not change the surface and cross-section morphology of the coating. Figure 5d–f shows the cross-section morphology of the coating oxidized at 800 °C. It is obvious that some oxides were generated on the coating surface. The EDS results of spot 5 in Table 4 demonstrated that most of the oxides were Al_2_O_3_, but they were not continuous. After oxidation for 50 h and 100 h, an off-white aluminum-poor band (spot 6) appeared under the oxides on the coating surface, and the elements diffused inside the coating to form dark gray (Al-Ni-rich, spot 7) and light gray (Fe-Cr-rich, spot 8) lines. From spot 9, it could be determined that most of the oxides in the lamellar gap of the coating were Al_2_O_3_. The cross-section morphology of the coating (see Figure 5g–i) showed that a continuous dense oxide layer was generated on the surface of the coating, and it was layered with a rather thick aluminum-poor layer underneath. After 100 h of oxidation, the thickness of the oxide layer and the aluminum-poor layer were about 2 μm and about 4 μm, respectively, and the internal diffusion of the coating was severe. The cross-section morphology of the coating (see Figure 5j–l) shows that the oxide layer had become considerably thicker, and its thickness increased with the oxidation time. It was about 4 μm thick after oxidation at 1000 °C for 100 h. It can be seen from spots 10–12 in Table 4 that the content of Al element decreased gradually with the deepening of the coating. Spot 10 is on the surface of the oxide film, which was mainly composed of α-Al_2_O_3_ and mixed oxides, and Spot 12 shows dense α-Al_2_O_3_ oxides rich in Al and O. From Figure 5l it was found that the aluminum-poor layer was thicker, up to 13 μm. Spot 13 shows that the content of elements in the aluminum-poor layer, and Al and O were reduced. Al is the phase stabilizer of BCC, and the rapid depletion of Al gradually transforms the matrix under the aluminum-poor layer into FCC phases [37]. The phases after 100 h of oxidation at different temperatures were high, consistent with the XRD patterns in Figure 3a. The FCC phases appeared due to the diffusion of Al, showing the BCC–FCC transition.

## 4. Discussions

In order to better understand the oxidation mechanism of the coating, EDS mapping was used to study the atomic diffusion and rearrangement that occurs because of high-temperature oxidation. Figure 6 is the element mapping of a section of AlCoCrFeNi coating after oxidation for 100 h. All components of the coating were uniformly distributed after oxidation at 650 °C for 100 h (see Figure 6a), and no oxide was generated on the coating surface. After oxidation at 800 °C for 100 h, it can be seen that many oxides had formed on the coating surface (see Figure 6b). The oxides were discontinuous and did not form a continuous oxide layer. These oxides were mixed oxides mainly consisting of Al_2_O_3_, and oxides mainly composed of Al_2_O_3_ will also be formed at the pores inside the coating. A coating oxidized at 900 °C for 100 h showed a continuous oxide layer (see Figure 6c), below which an aluminum-poor band was clearly visible, which is the FCC phase. In the coating oxidized at 1000 °C for 100 h (see Figure 6d), the oxide layer and the aluminum-poor layer became significantly thicker, and the oxide layer consisted of dense Al_2_O_3_ and mixed oxides. This was highly consistent with the XRD images in Figure 3a, which showed the gradual increase in Al_2_O_3_ and the BCC–FCC transition. Al_2_O_3_ gradually increased with the temperatures as Al diffused to the coating surface to form Al_2_O_3_. In addition, the BCC phases under the oxide layer gradually transformed into FCC phases. Only the aluminum-poor layer could be detected because of the limited detection depth of XRD images. Therefore, after 100 h of oxidation at 900 °C, the BCC phase disappeared completely.

At the initial stage of oxidation, the coating surface has sufficient O, thus all elements can undergo an oxidation reaction, generating mixed oxides upon it, as shown in Table 5 [38,39,40].

Al_2_O_3_ has the highest negative value of Gibbs free energy and is most likely to form. In addition, the oxygen partial pressure required for its formation is much lower than that for the oxides formed by Cr, Fe, Co, and Ni; therefore, Al_2_O_3_ was identified as the main oxide. After many of the elements on the coating surface are consumed, each element diffuses upward. Aluminum diffuses so much that it was found to be absent in the area below the oxide layer. After the formation of a dense oxide layer, the partial pressure of oxygen underneath it decreases dramatically. Except for Al_2_O_3_, other oxides are difficult to form, so a dense Al_2_O_3_ layer will be formed under the oxide layer.

Figure 7 shows the line scanning of the coating after being oxidized 1000 °C for 100 h. The content of O gradually became higher as it moved towards the coating surface, and that of Al at the bottom of the oxide layer showed a peak, indicating that at the bottom of the oxide layer was Al_2_O_3_ and at the upper part were mixed oxides.

Figure 8 shows the schematic diagram of coating surface oxidation after high temperature oxidation. From its cross section, serious diffusion was found in the coating interior. In order to investigate this, SEM morphology analysis was performed on the coating interior, as shown in Figure 9. After spraying, the coating had a mixed structure of BCC/B2. After oxidation at 650 °C (see Figure 9a–c), white spots gradually appeared in the coating interior, and they were A2 phase (disordered BCC phases) rich in Fe-Cr. As the oxidation temperatures and time increased, the white dots gradually turned into long rods embedded in the grey matrix, which were the Al-Ni rich ordered BCC phase (B2 phase). After oxidation at 800 °C (see Figure 9d–f), the long rods gradually grow up and connect with each other to form a polygonal rod-like structure. With the increase in oxidation time, the volume fraction of bright white A2 phases and gray B2 phases tended to be the same, and their distribution was uniform. With the increase in oxidation temperatures, A2 phases, which gradually turned into the matrix, were completely linked and wrapped around B2 phases, which gradually shrank and formed blocks. This is similar to the results found by Wang et al. [38,41]. As shown in Figure 10, the total volume fraction of the A2 phase inside the coating after oxidation was calculated at different temperatures and times.

As a result of oxidation, the diffusion and rearrangement of atoms occurs, leading to phase transition in the AlCoCrFeNi coating. Therefore, in order to further study the mechanism of diffusion, EDS element content analysis was carried out on the A2 phase and B2 phase after 100 h oxidation at different temperatures. The results are shown in Figure 11. The EDS results of B2 (see Figure 11a) showed that the content of Al and Ni in B2 phases increased significantly with oxidation temperatures; that of Co remained almost unchanged; that of Fe and Cr decreased slightly, which perhaps can be explained by the fact that the diffusion of Al and Ni changed the relative content of the other elements [38,42,43,44]. The EDS results of A2 phases (see Figure 11b) showed that the content of other elements was almost the same except for Al, which decreased significantly with increasing temperature. Al and Ni atoms have a high affinity due to the negative enthalpy of mixing (−22 kJ/mol) [45]. Thus they are more likely to be combined to form B2 phases. Different atomic sizes will lead to a different solid solution, which can be explained by Equation (1):(1)$$\;\;\;\delta =100\%\sqrt{\sum_{i}x_{i}+{\left(1-r_{i}/\bar{r}\right)}^{2}},\bar{r}=\sum_{i}x_{i}x_{i}$$

In this equation, δ is the atomic size difference of the alloy, and $r_{i}$ is the atomic radius of element i [46].

The larger the atomic size difference of the high-entropy alloys is, the more likely it is to have the BCC structure (relative to the FCC structure). The relatively loose BCC structure provides more space for atomic substitution than the tightly packed FCC structure, resulting in less lattice distortion in solution. Therefore, the BCC structure was easier to form in the Fe-Co-Ni-Cr system than the FCC. Al and Ni atoms were the most significantly different in terms of atomic size difference, which will become more obvious. Thus Al was regarded as an unstable agent for FCC. During the oxidation process, Al and Ni aggregate to form the ordered B2 structure, while the remaining components form the disordered BCC structure (A2, rich in Fe-Cr). Among the Al-3d transition metal interactions analyzed by Tang et al. using the phase diagram calculation [37], the B2-type Al-Ni was the most strongly bonded, with an enthalpy of 43 kJ/mol at 800 °C, while Cr Al and Fe Al solid solutions have positive enthalpies in BCC or FCC structures. At high temperatures, Al was more compatible with Ni and repelled Fe and Cr, which was consistent with the results offered by this study. In addition, Co diffuses slowly in the AlCoCrFeNi high-entropy alloys [47], which explains why the content of Co remains almost unchanged after oxidation. Based on the description above, the internal microstructure evolution of the oxidized coatings is summarized in Figure 12.

## 5. Conclusions

In this work, AlCoCrFeNi high-entropy alloy coatings were prepared by plasma spraying and were oxidized at 650–1000 °C for different time frames. The following conclusions were drawn after studying their oxidation behavior:(1)At the initial stage of oxidation, mixed oxides were formed on the surface of the HECs. After that, they formed a continuous oxide layer; the oxygen partial pressure under the layer dropped significantly, so only Al_2_O_3_ could be formed. Al in the coating diffused to the layer to form the Al_2_O_3_ layer. As a result, the oxide layer was laminated, consisting of mixed oxides on the surface layer and Al_2_O_3_ on the bottom layer. The HECs under the oxide layer experienced BCC-FCC transition because of the mobility of Al.(2)During the oxidation process, the coating gradually diffused from BCC/B2 mixed structure into B2 phases enriched with Al, Ni and A2 phases enriched with Fe, Cr. The A2 phases looked like rods at the beginning, but then they grew and linked each other to wrap the B2 phases.

## Figures and Tables

**Figure 1 materials-17-00550-f001:**
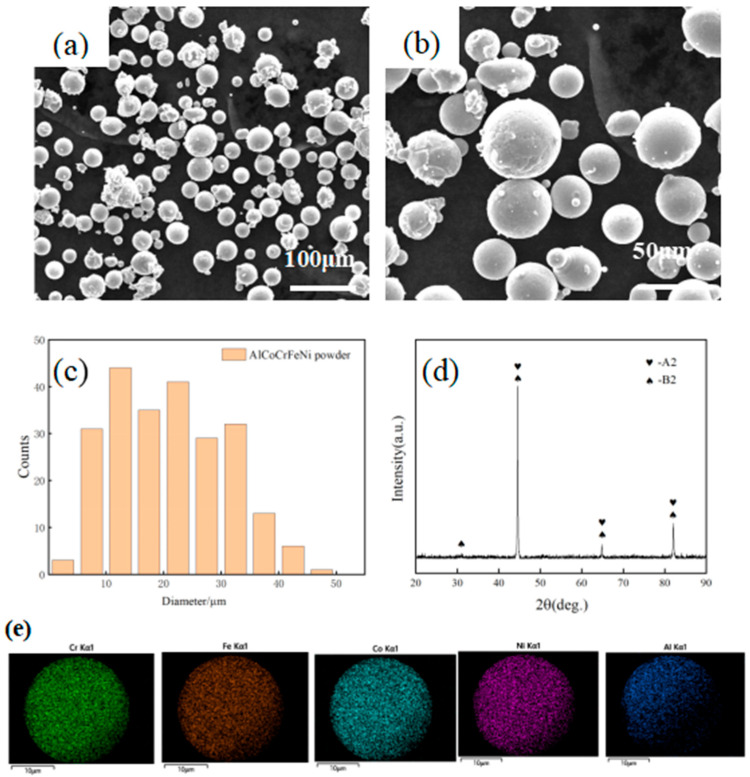
SEM morphology (**a**) and (**b**) particle size distribution; (**c**) XRD spectrum (**d**) of atomized AlCoCrFeNi powders; (**e**) mapping of AlCoCrFeNi powder.

**Figure 2 materials-17-00550-f002:**
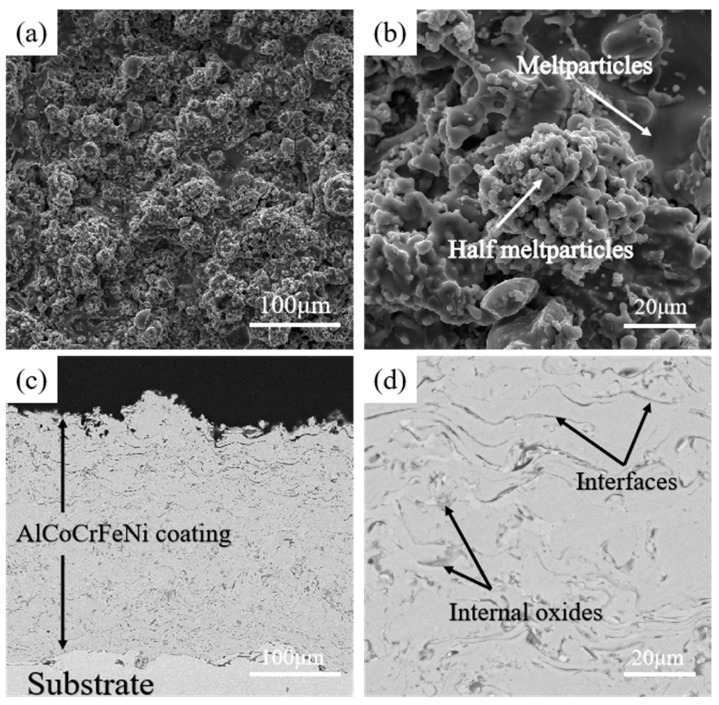
SEM morphology of surface (**a**) and (**b**) and section (**c**) and (**d**) of sprayed AlCoCrFeNi coating.

**Figure 3 materials-17-00550-f003:**
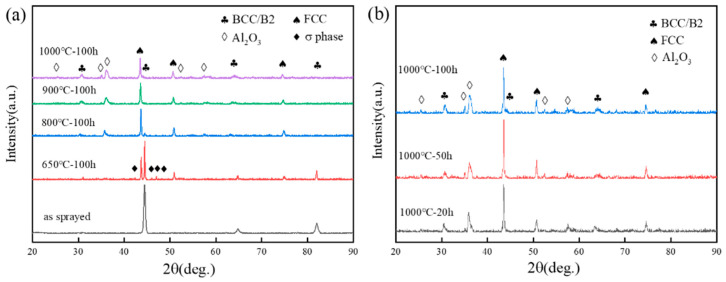
XRD images on the surface of AlCoCrFeNi coating: (**a**) the sprayed coating and the coating oxidized at 1000 °C, 900 °C, 800 °C and 650 °C for 100 h. (**b**) The XRD images of the coating oxidized at 1000 °C for 20 h, 50 h and 100 h.

**Figure 4 materials-17-00550-f004:**
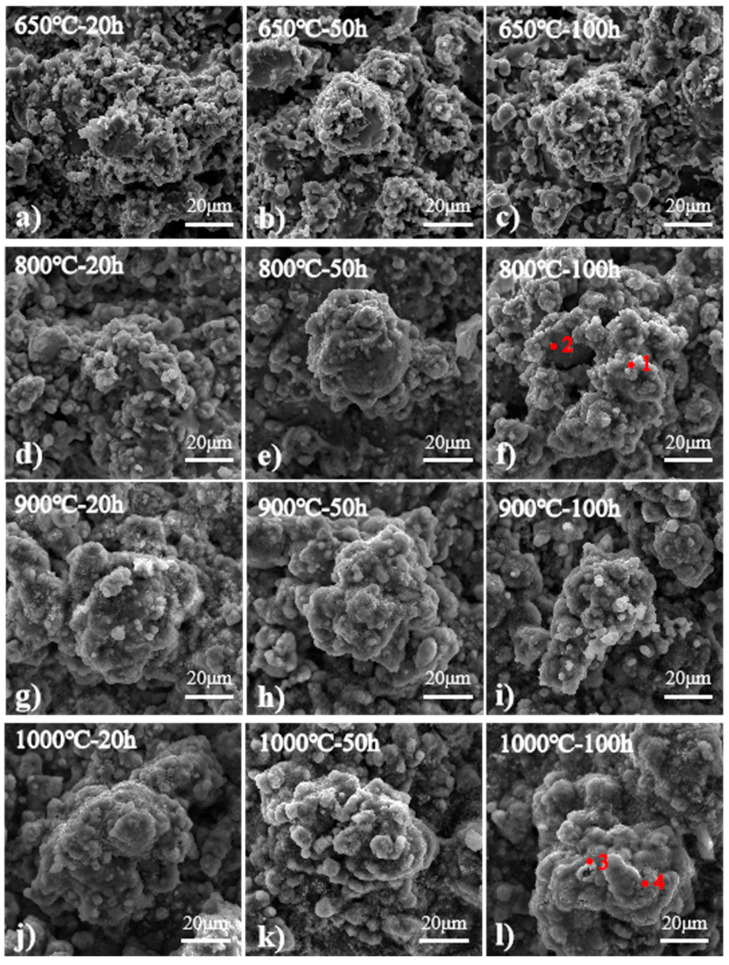
Surface morphology of coating after oxidation at different temperatures and for different times: (**a**) oxidation at 650 °C for 20 h, (**b**) oxidation at 650 °C for 50 h, (**c**) oxidation at 650 °C for 100 h, (**d**) oxidation at 800 °C for 20 h, (**e**) oxidation at 800 °C for 50 h, (**f**) oxidation at 800 °C for 100 h, (**g**) oxidation at 900 °C for 20 h, (**h**) oxidation at 900 °C for 50 h, (**i**) oxidation at 900 °C for 100 h, (**j**) oxidation at 1000 °C for 20 h, (**k**) oxidation at 1000 °C for 50 h, (**l**) oxidation at 1000 °C for 100 h.

**Figure 5 materials-17-00550-f005:**
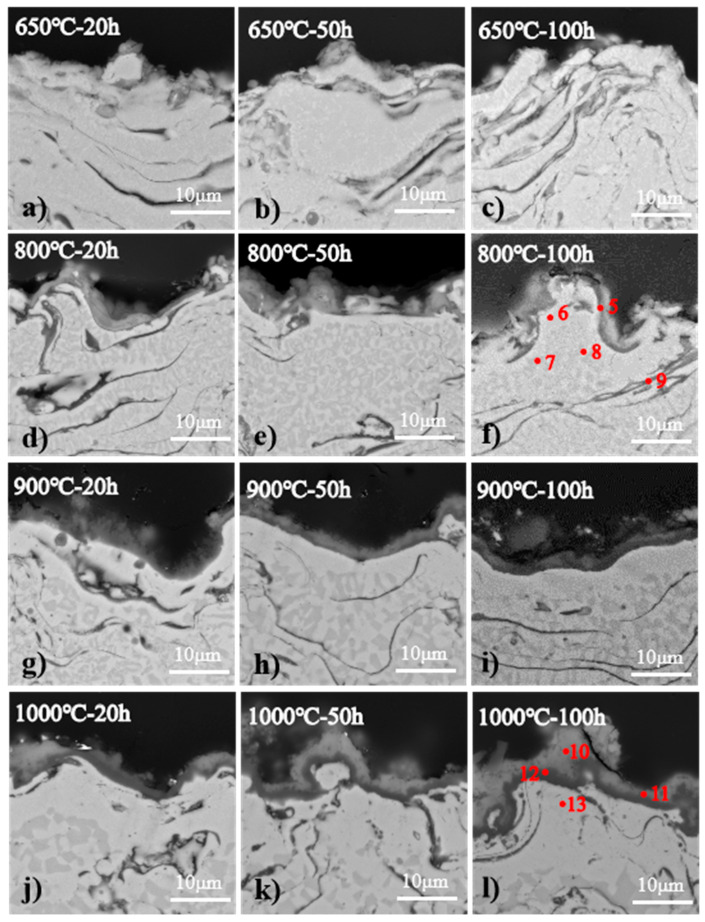
Cross-sectional morphology of coating after oxidation at different temperatures and different times: (**a**) oxidation at 650 °C for 20 h, (**b**) oxidation at 650 °C for 50 h, (**c**) oxidation at 650 °C for 100 h, (**d**) oxidation at 800 °C for 20 h, (**e**) oxidation at 800 °C for 50 h, (**f**) oxidation at 800 °C for 100 h, (**g**) oxidation at 900 °C for 20 h, (**h**) oxidation at 900 °C for 50 h, (**i**) oxidation at 900 °C for 100 h, (**j**) oxidation at 1000 °C for 20 h, (**k**) oxidation at 1000 °C for 50 h, (**l**) oxidation at 1000 °C for 100 h.

**Figure 6 materials-17-00550-f006:**
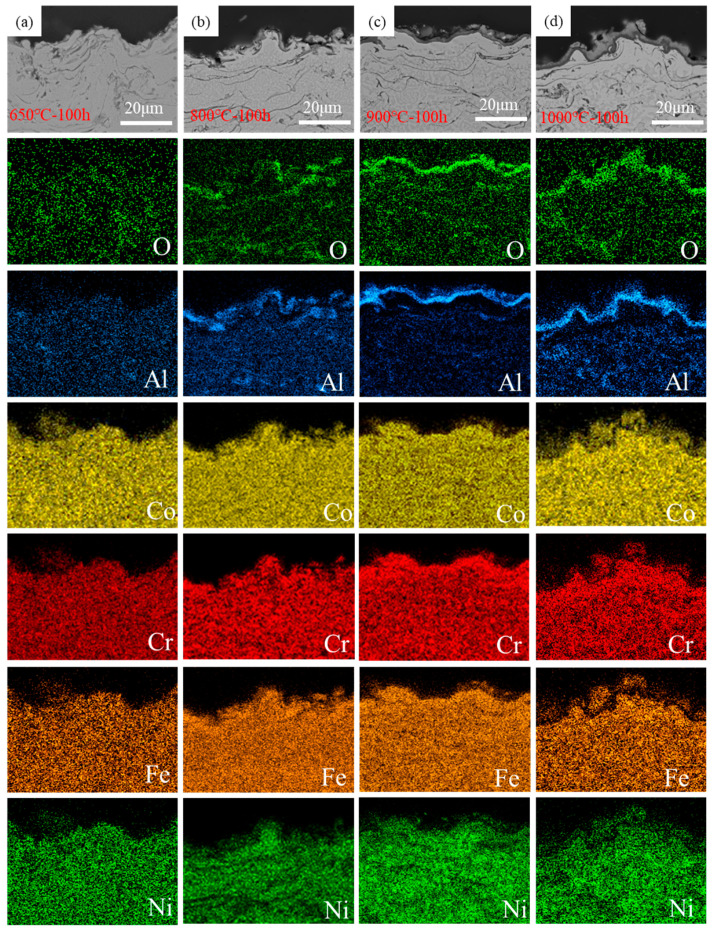
Element mappings of AlCoCrFeNi coating cross section after 100 h oxidation: (**a**) oxidation at 650 °C, (**b**) oxidation at 800 °C, (**c**) oxidation at 900 °C, (**d**) oxidation at 1000 °C.

**Figure 7 materials-17-00550-f007:**
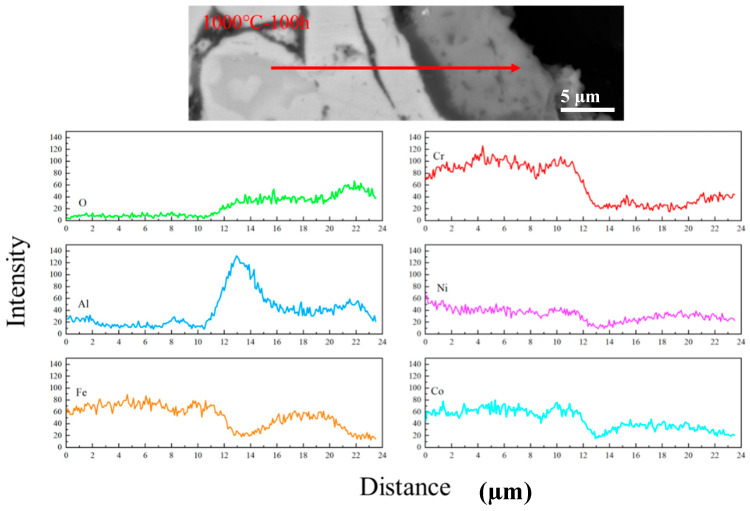
Analysis of element concentration of AlCoCrFeNi coating after oxidation at 1000 °C for 100 h.

**Figure 8 materials-17-00550-f008:**
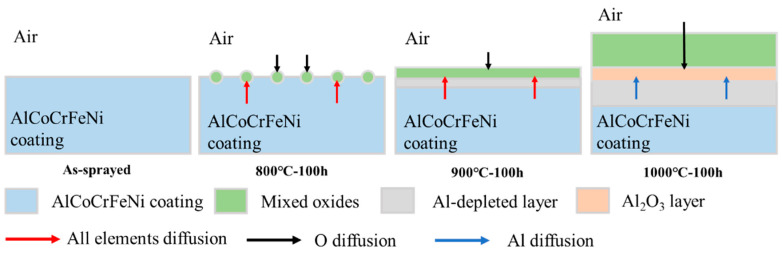
Schematic diagram of coating surface oxidation after high temperature oxidation.

**Figure 9 materials-17-00550-f009:**
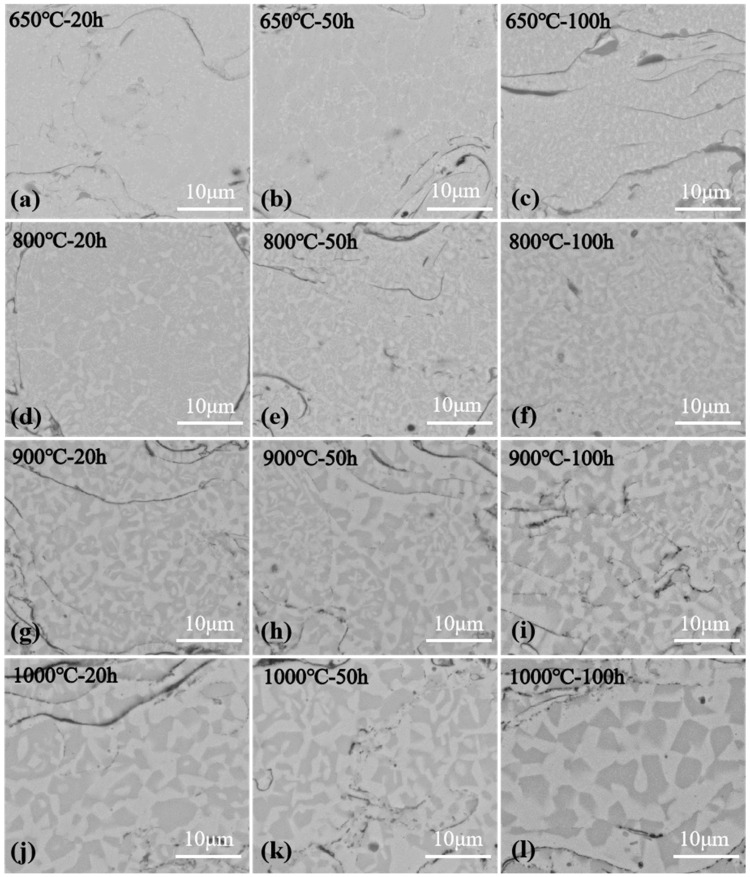
BSE images of cross-sectional morphology of plasma sprayed AlCoCrFeNi coating after oxidation at different temperatures and different times: (**a**) oxidation at 650 °C for 20 h, (**b**) oxidation at 650 °C for 50 h, (**c**) oxidation at 650 °C for 100 h, (**d**) oxidation at 800 °C for 20 h, (**e**) oxidation at 800 °C for 50 h, (**f**) oxidation at 800 °C for 100 h, (**g**) oxidation at 900 °C for 20 h, (**h**) oxidation at 900 °C for 50 h, (**i**) oxidation at 900 °C for 100 h, (**j**) oxidation at 1000 °C for 20 h, (**k**) oxidation at 1000 °C for 50 h, (**l**) oxidation at 1000 °C for 100 h.

**Figure 10 materials-17-00550-f010:**
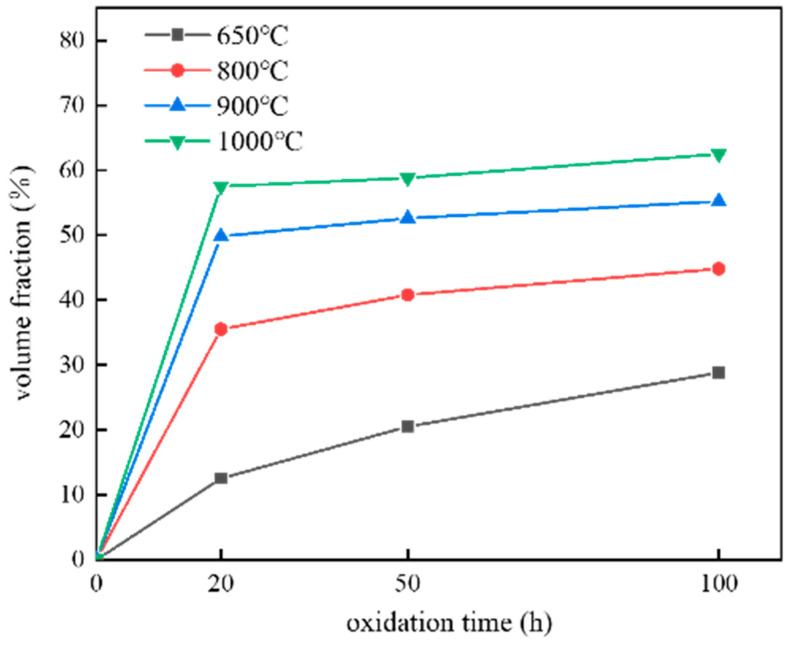
Volume fraction of A2 phase inside the oxidized coating.

**Figure 11 materials-17-00550-f011:**
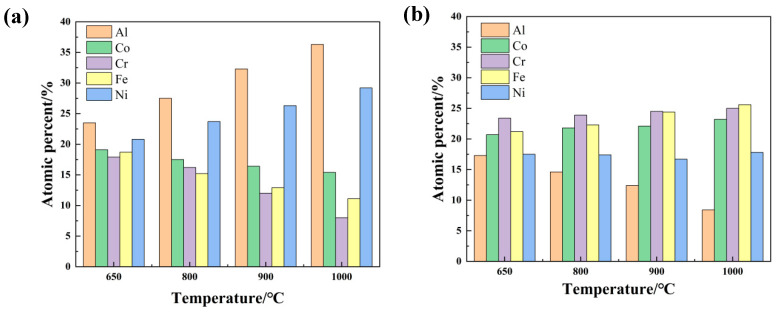
Results of EDS content analysis of coatings after oxidation at different temperatures for 100h (**a**) EDS results of B2 (**b**) EDS results of A2.

**Figure 12 materials-17-00550-f012:**
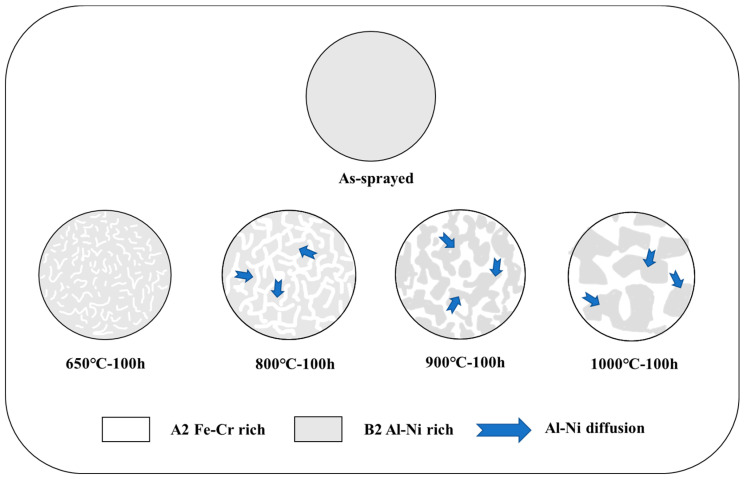
Schematic diagram of internal microstructure evolution of AlCoCrFeNi coating after oxidation.

**Table 1 materials-17-00550-t001:** Plasma spraying parameters of AlCoCrFeNi coating.

Process Parameter	Unit	AlCoCrFeNi Coating
Spraying current	A	550
Spraying voltage	V	62
Spraying power	kW	34.1
Standoff distance	mm	110
Scanning rate of plasma torch	mm	150

**Table 2 materials-17-00550-t002:** Atomic percentage of each element in AlCoCrFeNi powder.

Element	Al	Co	Cr	Fe	Ni
At%	23.8	19.0	19.8	19.1	18.3

**Table 3 materials-17-00550-t003:** EDS results for the labeled regions in Figure 4.

Spot	Composition (At %)					
	Al	Co	Cr	Fe	Ni	O
1	13.6	5.5	5.9	6.1	4.4	64.5
2	5.7	6.9	7.7	8.3	5.7	65.7
3	5.3	3.4	4.7	8.0	4.0	74.6
4	4.4	4.0	4.1	10.7	4.1	72.6

**Table 4 materials-17-00550-t004:** EDS results for the labeled regions in Figure 5.

Spot	Composition (At %)					
	Al	Co	Cr	Fe	Ni	O
5	21.4	6.9	7.8	6.9	5.9	55.1
6	5.8	23.9	24.6	25.7	18.3	1.8
7	30.7	15.8	10.4	13.7	28.8	0.5
8	13.3	20.3	20.3	22.3	15.1	8.3
9	26.9	8.8	11	9.6	7.8	35.9
10	14.6	6.0	8.2	4.5	3.9	62.7
11	30.5	2.7	2.7	1.7	2.2	60.2
12	35.9	1.2	2.3	1.2	1.0	58.4
13	8.2	21.6	25.4	24.5	15.4	4.9

**Table 5 materials-17-00550-t005:** Gibbs free energy to form various oxides (Gf: kj/mol) [9].

Al_2_O_3_	Cr_2_O_3_	FeO	Fe_3_O_4_	Fe_2_O_3_	CoO	NiO
880.35	563.55	398.45	377.65	353.45	310.00	277.45

## Data Availability

The authors declare that [the/all other] data supporting the findings of this study are available within the article.

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
