# Peer review of "Initial Oxidation Behavior of AlCoCrFeNi High-Entropy Coating Produced by Atmospheric Plasma Spraying in the Range of 650 °C to 1000 °C"

_materials, 2024, doi:10.3390/ma17030550_

Round 1
Reviewer 1 Report
Comments and Suggestions for Authors
In this work, the authors have prepared AlCo-CrFeNi high-entropy coatings (HECs) by using atmospheric plasma spraying (APS). The coating is then oxidized from 650 °C to 1000 °C to study the mechanism of the oxide layer formation and internal phase transition behavior. The article is logical and well organized. All the hypothesis and claims are well supported with the design of the experiments and instrumental analysis. Additionally, the scope of work covered is appropriate and likely to appeal to the MDPI’s audience. The reviewer suggests acceptance of this work in its current format.

Author Response
We would like to thank to Reviewer 1 for acknowledging our work.
Reviewer 2 Report
Comments and Suggestions for Authors
The author focuses on the research of high-temperature oxidation and phase transformation of AlCoCrFeNi coating. The article is well-written and it can be published after minor questions are confirmed.
1/ author mentioned, “Fig. 3b shows that the phase composition of the coating after spraying is consistent with that of the powder, indicating that the spraying process has not changed the phase structure of the coating…”. How about true element compositions of the coating film? Fig. 3b is mentioned first. I think you can reverse Fig. 3a and b.
2/ You explained that “As Al diffuses to the coating surface to form Al2O3, Al2O3 gradually increases with the temperatures”. In Fig. 6d and 7(Fe), besides the proof of Al, the appearance of Fe at the outside layer also are observed. Why do you only focus on the performance of Al2O3? How about the role of other oxides?
3/It is interesting to observe the separation of Al-rich layer and outside mixed oxide layers (Fig. 5j-l). Fig. 12, the author confirms the AL-Ni diffusion and their bond is strong enough to form B2 structure. For this result, why the concentration of Al2O3 is higher than Ni oxide at outside oxide layer? Should the amount of their oxides be similar?
4/Does element diffusion and oxidation layer cause a decrease of the mechanical of coating layer or cracking?
5/In the introduction part, the author did not mention the purpose of research on the formation and evolution process of oxide. How does it impact the mechanical properties and applicability of coating?
6/ Please check English spelling again.
Comments on the Quality of English Language
English language is well written, however, authors should check carefully minor mistakes before publication.
Author Response
We would like to thank to Reviewer 2 for his valuable comments.

Reviewer 3 Report
Comments and Suggestions for Authors
This paper has investigated the oxidation behavior of AlCoCrFeNi high-entropy coating produced by atmospheric plasma spraying at the range of 650 ℃ to 1000 ℃. However, there are some issues that need to be addressed before the manuscript can be accepted by the journal.
1) The language of the paper requires extensive corrections and polishing, as the reviewer found many grammatical and syntax errors.
2) The authors should provide a more comprehensive discussion about the different aspects and properties of high entropy alloys.
3) It is suggested that EDS elemental mapping of the gas atomized high entropy powders be added to show the distribution of elements.
4) Please calculate the crystallite size of the powders as well as the volume fraction of the phases based on the XRD results. Below is a reference to help with this process: Thermal Stability of Nano-Hydroxyapatite Synthesized via Mechanochemical Treatment, Arabian Journal for Science and Engineering volume 42, pages4401–4408 (2017).
5) More in depth discussion regarding to mechanism of formation of Al2O3 and sigma phases after oxidation is needed.
6) What is the effect of different atomic sizes parameter on phase formation or phase transformation during the oxidation? Below is a reference that may assist with this process: - Hierarchically activated deformation mechanisms to form ultra-fine grain microstructure in carbon containing FeMnCoCr twinning induced plasticity high entropy alloy, Materials science and engineering A, Volume 824, 2021, 14180

Comments on the Quality of English Language
The language of the paper requires extensive corrections and polishing, as the reviewer found many grammatical and syntax errors.
Author Response
We would like to thank to Reviewer 3 for his valuable comments.

Round 2
Reviewer 3 Report
Comments and Suggestions for Authors
The authors responded to my comments so it is suggested to publish paper.
Author Response
Thank you for your approval. We will continue to explore and modify some of the issues in the article.